# UniTransfer: Video Concept Transfer via Progressive Spatial and Timestep Decomposition

**Guojun Lei**[1,*] **Rong Zhang**[2,*] **Tianhang Liu**[1] **Hong Li**[4]
**Zhiyuan Ma**[3,†] **Chi Wang**[5] **Weiwei Xu**[1,†]
[1] State Key Lab of CAD&CG, Zhejiang University, [2] Zhejiang Gongshang University,
[3] Huazhong University of Science and Technology, [4] Beihang University,
[5] Ningbo Global Innovation Center, Zhejiang University
guojunlei@zju.edu.cn, zhangrong@zjgsu.edu.cn, link0502@buaa.edu.cn, tianhang@zju.edu.cn,
mzyth@hust.edu.cn, wangchi1995@zju.edu.cn, xww@cad.zju.edu.cn
[Web Page]

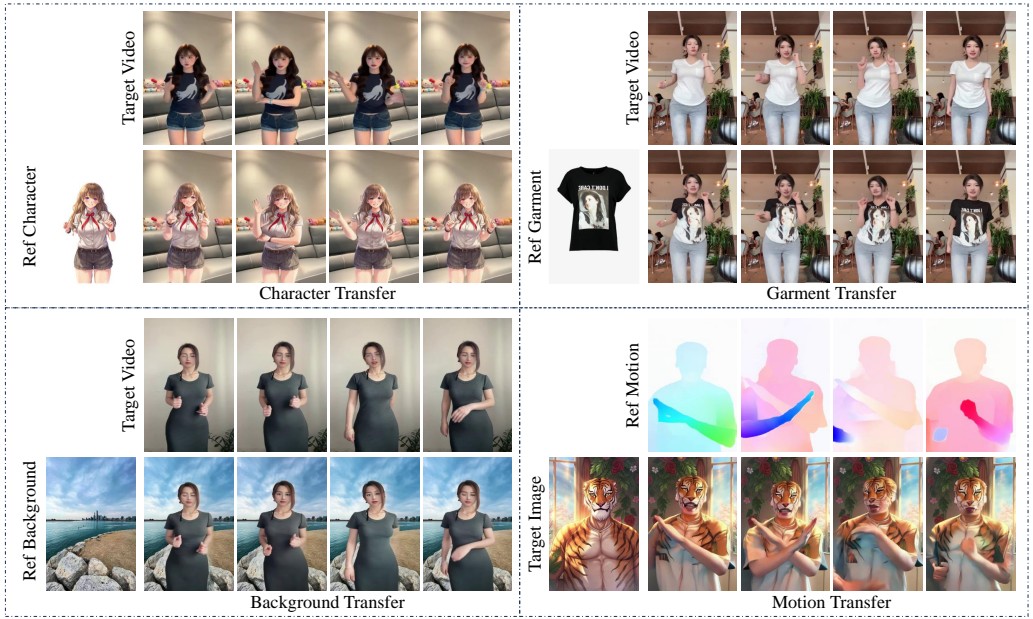

Figure 1: Results of our UniTransfer. These qualitative results exhibit the superior performance of our approach in transferring various reference components, including *characters*, *garments*, *backgrounds*, and *motions*, to synthesize the new target videos.

## Abstract

Recent advancements in video generation models have enabled the creation of diverse and realistic videos, with promising applications in advertising and film production. However, as one of the essential tasks of video generation models, video concept transfer remains significantly challenging. Existing methods generally model video as an entirety, leading to limited flexibility and precision when solely editing specific regions or concepts. To mitigate this dilemma, we propose a novel architecture UniTransfer, which introduces both spatial and diffusion timestep decomposition in a progressive paradigm, achieving precise and controllable video concept transfer. Specifically, in terms of spatial decomposition, we decouple videos into three key components: the

---

*Equal contributions.

†Corresponding authors.

39th Conference on Neural Information Processing Systems (NeurIPS 2025).

foreground subject, the background, and the motion flow. Building upon this decomposed formulation, we further introduce a dual-to-single-stream DiT-based architecture for supporting fine-grained control over different components in the videos. We also introduce a self-supervised pretraining strategy based on random masking to enhance the decomposed representation learning from large-scale unlabeled video data. Inspired by the Chain-of-Thought reasoning paradigm, we further revisit the denoising diffusion process and propose a Chain-of-Prompt (CoP) mechanism to achieve the timestep decomposition. We decompose the denoising process into three stages of different granularity and leverage large language models (LLMs) for stage-specific instructions to guide the generation progressively. We also curate an animal-centric video dataset called OpenAnimal to facilitate the advancement and benchmarking of research in video concept transfer. Extensive experiments demonstrate that our method achieves high-quality and controllable video concept transfer across diverse reference images and scenes, surpassing existing baselines in both visual fidelity and editability. Web Page: https://yu-shaonian.github.io/UniTransfer-Web/

# 1 Introduction

Recent years, the rapid advancement of generative AI technologies [13, 36, 33] has greatly reformed the community of video editing, opening up new potentials for diverse, fine-grained, and user-controllable content manipulation tasks. Video Concept Transfer (VCT) is one of the most important downstream tasks in video editing, aiming to substitute various user-specified target concepts in a video, such as objects, characters, backgrounds, or subject motions, to enable personalized content manipulation. Its applications span across diverse areas such as film production, game development, virtual reality, *etc.* , drawing increasing attention from both academia and industry.

Despite its potential, achieving high-quality video concept transfer is still challenging as it requires seamless integration of different components, preservation of the identity of the target object, and visual fidelity of the generated videos. Some recent approaches rely on text-based guidance to control the transfer [10, 4, 41]. However, the inherent ambiguity of natural language descriptions often results in imprecise manipulation of object attributes and behaviors, restricting their application scenarios. In contrast, image-guided methods offer a more accurate way to encode the appearance and identity of target objects. For example, methods like AnimateAnyone2[15] and MIMO [29] have demonstrated the ability to replace human subjects in videos with specific reference images. However, these existing approaches mainly focus on human transfer and struggle to generalize to broader editing tasks [25] involving arbitrary objects, backgrounds, garments or motion patterns. VideoSwap [11] and AnyV2V [20] rely on personalized modeling or image editing techniques to address this dilemma. But the ability of their foundation models limits their scalability and flexibility in complex videos.

This paper focuses on image-based video concept transfer, including animals, characters, backgrounds, and motions. This task inherently involves manipulations and integrations of different components within a video, which often exhibit substantial variation in terms of visual appearance, motion pattern, or semantic attributes. However, most existing approaches [23, 41, 11, 20, 28] generally model the video as a unified whole without considering the heterogeneous nature of its constituents, which may introduce undesired artifacts. MIMO [29] introduced spatial decomposed modeling to VCT by decomposing a video into three predefined components: human, scene, and occlusion. It helps improve the quality of video character replacement. However, MIMO only decomposes videos in the spatial dimension, which is not sufficient for high-quality video generation, as it takes all the timesteps in the denoising process equally. Motivated by ProSpect [53], diffusion models actually generate images in the progressive order of *"layout → content → texture"*, and this work also exhibits that different stages in the diffusion models require guidance at different granularities.

In this work, we propose UniTransfer, a Diffusion Transformer (DiT) based video concept transfer framework via progressive decomposition of both the spatial dimension and the denoising process. In the spatial dimension, we decompose videos into three core components: foreground, background, and motion dynamics, enabling our model to adapt to general concept transfer flexibly. To achieve this, we allow the model to learn the decomposition from coarse foreground masks to detailed ones. Specifically, we first introduce a random masking-based self-supervised pretraining strategy to

strengthen the decomposed representation learning, which enables the model to capture disentangled features without requiring fine-grained annotations. Then we design a dual-to-single-stream Dit architecture to realize further decomposition with delicate semantic annotations. In this stage, individual branches are responsible for encoding different video components, and their features are later integrated into a unified representation through a single-stream network. This design enhances the model's capacity to manipulate different objects and maintain the temporal consistency.

In the denoising process, we decompose the timestep into coarse-grained, mid-grained, and fine-grained stages instead of modeling it equally. Inspired by the Chain-of-Thought (CoT) strategy, we develop a Chain-of-Prompt (CoP) mechanism and leverage Large Language Models (LLMs) to produce hierarchical prompts at different granularities and utilize them to guide the generation, enabling progressive refinement from noise to detailed textures. The main contributions are summarized as follows:

- We propose a DiT-based image-guided video concept transfer framework UniTransfer, which incorporates progressive spatial and timestep decomposition.

- We introduce a self-supervised pretraining strategy based on randomized masking to enhance the disentangled representation learning and design a dual-to-single-stream architecture to achieve spatial decomposition.

- We further introduce an LLMs-guided chain-of-prompt mechanism to achieve the timestep decomposition. This progressive prompting strategy guides the generation process with stage-specific instructions, improving the VCT generation quality.

- We collect an animal-centric video dataset called OpenAnimal to facilitate the training and benchmarking of research in video concept transfer. Extensive experiments demonstrate that our method outperforms state-of-the-art methods in various video concept transfer scenarios.

## 2 Related Work

**Text-driven Video Editing.** Video editing has witnessed remarkable advances in conditional content synthesis and manipulation through diffusion-based architectures [27]. Recently, video editing typically relies on textual prompts to control object attributes or behaviors [10, 4, 1]. Video-P2P [23, 32] achieves preservation of motion dynamics through attention modulation. RF-Edit [41] preserves structural integrity and temporal consistency by rectified flow ODE solving with reduced error. MotionDirector [55] requires separate training for each motion type. When adapting to new scenarios, it only supports text-guided scene transitions, which significantly limits its flexibility. VMC [18] also needs seperate training for different scenes. However, due to the inherent ambiguity and underspecification of natural language, such methods often struggle to achieve fine-grained and accurate control over video content.

**Image-guided Video Editing.** To overcome these limitations, researchers introduce additional reference images to guide the editing process of the video subject. MIMO [29] and MovieCharacter [34] decompose the video into elements such as foregrounds and preprocessed backgrounds, and perform character transfer through video composition techniques. AnimateAnyone2 [15] proposes a framework to animate characters while considering environmental affordances. Despite their effectiveness, they are designed for character video editing, which limits the application scenarios.

In contrast, VideoSwap [11] pioneers a more general approach to image-guided concept transfer through semantic correspondence learning, enabling more versatile and accurate video edits beyond character animation. Yet critically, its reliance on multi-image concept anchoring introduces semantic abstraction, achieving category-level consistency (e.g., kitten, airplane) but failing to preserve instance-specific attributes. AnyV2V [20] leverages arbitrary image editing tools [3, 42, 7] to modify the first frame of a video and then propagates the modifications to subsequent frames, enabling instance-based object-driven editing and identity manipulation. However, the two-stage pipeline heavily depends on the results of off-the-shelf image editing techniques, and the temporal consistency can not be guaranteed. To address these limitations, we propose a novel framework that does not rely on external image editing tools. Instead, our method decomposes the video into three disentangled components: foreground, background, and motion. Integrating with a carefully-tailored network architecture in conjunction with a large language model (LLM)-based chain-of-thought reasoning mechanism, our method can achieve precise and flexible video concept transfer.

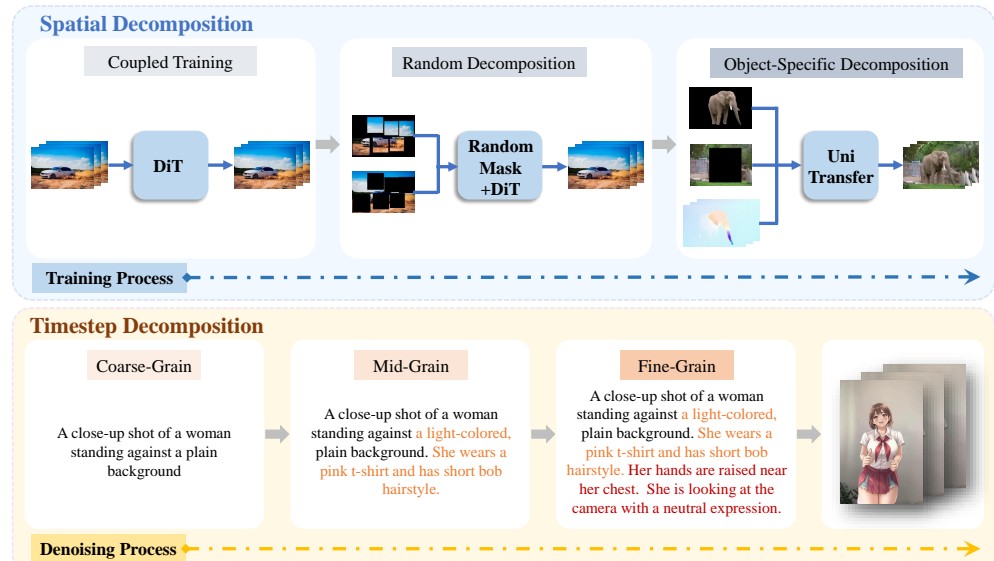

Figure 2: Our progressive spatial and timestep decomposition modeling.

**Chain-of-Thought Prompting** [19] aims to substantially improve the reasoning capabilities of large language models (LLMs) [6]. It provides a framework for complex multi-step inference through explicit intermediate reasoning steps. The CoT concept evolves into CoX [44], where X denotes swappable nodes (e.g. intermediates [19, 56, 21, 40], augmentation [12, 49, 9, 52], feedback [46, 24, 2, 8], and models [47, 5, 45]) for task-specific adaptation. We further propose Chain-of-Prompt (CoP), which shifts the CoX paradigm to the iterative denoising steps of diffusion models. This systematic approach decomposes the video generation into sequential coarse-to-fine steps, where hierarchical prompt guidance progressively refines temporal features through successive diffusion iterations.

## 3 Method

In this section, we present UniTransfer, a DiT-based image-guided video concept transfer framework with progressive decomposed video modeling. Our goal is to generate high-quality videos with user-specified concepts in the referenced images, including objects, characters, animals, backgrounds, or motion dynamics. To achieve this, we decompose the video both in the spatial and the timestep dimension. The overview of the proposed framework is illustrated in Figure 2.

### 3.1 Spatial Decomposition

Existing video generation methods typically treat the video as a holistic entity and attempt to model it under the guidance of text prompts or reference images. Although such strategies are effective for general video synthesis, they fall short in the context of video concept transfer, which requires compositional control over different parts of the video, including foreground objects, background scenes, and motion dynamics. Encoding the entire video into a single latent space makes it difficult to independently manipulate these components during generation. As a result, existing image-based video object transfer methods are often limited to transferring only the main subject, instead of manipulating different components, including background appearance or motion dynamics.

To achieve more flexible and controllable video concept transfer, we propose to spatially disentangle the video generation process. Specifically, a video can be decomposed into three distinct components: the foreground $M$, the background $B$, and the corresponding motion flow $F$. In general, the foreground component may consist of different objects (e.g., characters, animals, or objects). Under this setting, we redefine the video denoising process as follows:

$$\mathcal{L}(\theta) = \mathbb{E}_{x_0, \epsilon, \mathcal{U}, t} \left[ \| \epsilon - \hat{\epsilon}_\theta \left( x_t, \tau, \mathcal{U}, t \right) \|_2^2 \right],$$
(1)

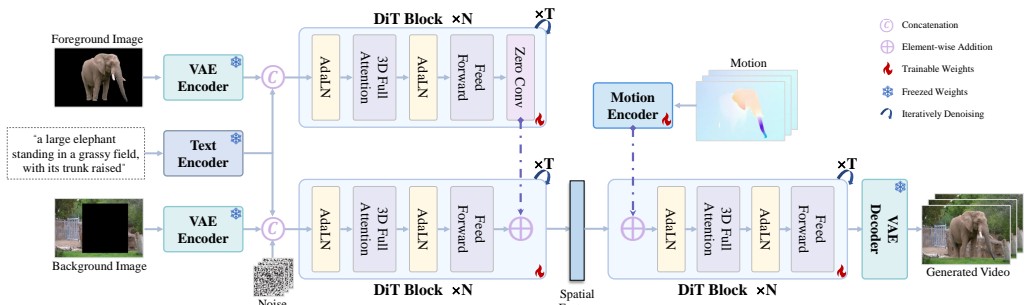

Figure 3: The architecture of our UniTransfer.

where $\tau$ is the condition of text prompts, $\mathcal{U} = (M, B, F)$. This decomposition enables us to treat each component independently and recombine them flexibly in the generation pipeline, laying the foundation for video concept transfer.

Building upon this decomposed formulation, we propose a progressive learning scheme that enables the model to effectively capture and utilize the decomposed components for video generation. To achieve this, we design two modules: (1) a random masking-based self-supervised learning mechanism to model the coarse relationships between the foregrounds and the backgrounds without delicate semantic annotations. (2) a carefully designed dual-to-single stream architecture, UniTransfer, tailored to learn detailed interactions of three different components. In the following, we will introduce them respectively.

### 3.1.1 The Architecture of UniTransfer

In this stage, we propose a DiT-based architecture named UniTransfer to decompose the video into three components: the foreground appearance, the background scene, and the motion dynamics to enable flexible and fine-grained control over VCT. Unlike previous approaches that treat the video as a monolithic entity, UniTransfer is designed to disentangle and recompose these components in a controllable diffusion-based generation process.

In the training process, given the input video $V$, we first randomly sample a single frame to serve as a reference image $I$. A pretrained semantic segmentation model is then utilized to predict a foreground mask $M \in \{0, 1\}^{H \times W}$, which partitions the reference frame into a foreground region $F = I \odot M$ and a background region $B = I \odot (1 - BBox(M))$, where $\odot$ represents element-wise multiplication and $BBox(M)$ means all the pixels in the bounding box are setting to 1. With the unaligned masks, we force the model to learn shape-agnostic interactions between the foreground and the background. Meanwhile, motion dynamics are extracted from the entire video using a pretrained optical flow model RAFT [39], which captures the temporal dynamics independent of appearance content. UniTransfer takes $F, B, O$ along with the video description $\tau$ as guidance and iteratively denoises from a randomly sampled Gaussian noise map to generate a new video $\hat{V}$. Figure 3 illustrates the framework of the network, which is in a dual-to-single-stream paradigm consisting of three collaborative branches: a foreground branch, a background branch, and a fusion branch.

The foreground and background branches are built based on the CogVideoX architecture[48]. Each of them is composed of an image VAE encoder to project $F$ and $B$ into latent codes $z_f$ and $z_b$, a text encoder to provide shared high-level text embedding $z_\tau$, and a stack of $N$ DiT blocks for temporal feature modeling. In the background branch, a random Gaussian noise vector $z_t$ of the timestep $t$ is concatenated with $z_b$ and $z_\tau$, and the combined representation is passed through the DiT blocks $h_b^i$, for $i = 1, \ldots, N$. In contrast, the foreground branch is treated as a conditional stream with no noise input. It adopts a symmetric structure and produces intermediate feature maps for each DiT block $h_f^i$, for $i = 1, \ldots, N$. Afterwards, we design a feature injection module that introduces foreground features into the background stream inspired by ControlNet [51]. Specifically, the foreground feature map is processed by a zero-initialized convolutional projection layer and then element-wise added to the corresponding layer in the background branch at each DiT block. It allows the model to inject spatially aligned foreground appearance into the generative path.

Moreover, the enhanced background features are combined with the motion dynamics in the fusion branch, which consists of a motion encoder to project the optical flow to a latent code $z_o$, and $N$

DiT blocks for further feature fusion. The flow encoder adopts four symmetrically arranged stages, aligning with the structure in 3D VAE encoder [48]. Inspired by Tora[54], we also design an adaptive norm layer before adding it to the dit block. To enable the model to handle the misalignments between the motion and the input images and improve the generalization ability, we inject random noise into the input optical flow. $z_o$ is fused with the background stream through element-wise addition in each fusion DiT block $h^i$, for $i = 1, \ldots, N$.. The model outputs the predicted noise at a given denoising timestep $t$ as follows:

$$\hat{\epsilon}_\theta = h[ZConv(h_f(z_f \odot z_\tau)) \oplus h_b(z_t \odot z_b \odot z_\tau) \oplus z_o] \tag{2}$$

where $ZConv$ represents the zero-initialized convolution. $\odot$ and $\oplus$ represent the concatenation operation and element-wise addition, respectively.

### 3.1.2   Self-Supervised Pre-Training via Random Masking

Learning robust video representations for concept-aware generation typically requires large-scale datasets with annotated semantic masks or motions. However, existing video datasets rarely provide sufficient annotations. Relying on small-scale annotated datasets is not enough to directly learn spatially decoupled representations for video concept transfer. To address this limitation, we follow LMD [26] to introduce a random masking strategy before training with fine-grained annotations. It leverages large-scale unlabeled video datasets for learning initial disentangled representations. In this stage, we only learn the foreground and background decomposition for coarse initialization. In this stage, the binary foreground mask $M \in \{0, 1\}^{H \times W}$ is generated through random masking to arbitrarily partition the reference image $I$ into two regions $F$ and $B$. The mask is initialized as an all-zero matrix and is iteratively updated by randomly drawing rectangles of random size and position on it. The pixels inside each rectangle are set to 1, and the process continues until the foreground coverage exceeds $50\%$ of the image area. This masking strategy ensures spatial diversity and balance between foreground and background regions. Then the reference frame is partitioned into a foreground region $F = I \odot M$ and a background region $B = I \odot (1 - Dilate(M))$, where $Dilate(M)$ represents the morphological dilation operation to enable the model to learn the boundary interactions. Then $F$ and $B$ are fed into the foreground branch and a background branch, which inject the features into the denoising network independently. This allows the model to learn how to reconstruct coherent video content from partial and spatially isolated cues, without requiring ground-truth foreground-background labels. Through extensive self-supervised training on large-scale data, our model acquires strong prior knowledge, enabling efficient adaptation to downstream tasks via precise supervised fine-tuning.

### 3.2   Timestep Decompostion

Revisiting the preliminary of diffusion models reveals another limitation in current video generation pipelines: the same conditioning prompt is applied across all timesteps during the denoising process. it is difficult to generate grained texture corresponding to the grained prompts at the initial stage. Inspired from ProSpect [53], the role of each timestep varies significantly in the generation process. In the early stages of denoising, the model focuses primarily on recovering the global structure and semantic layout of the video. In contrast, the later stages are responsible for learning fine-grained details such as textures, colors, and subtle appearance attributes. Applying a static, single-level prompt across all timesteps ignores this progression in modeling focus. For instance, using overly complex or fine-grained textual prompts during early timesteps may misguide the model, leading to the omission of some semantic attributes or misalignments between the generated content and the input prompt.

To address this, we introduce a timestep decomposition mechanism named Chain-of-Prompt (CoP) inspired by the Chain-of-Thought reasoning paradigm in large language models. Specifically, we decompose the denoising timesteps into three granularity levels—coarse, medium, and fine—and leverage a large language model (LLM) to automatically generate three levels of prompts that progressively reflect the abstraction and detail required at different denoising stages. These prompts are then injected into the diffusion model in a stage-aware manner, ensuring that early timesteps receive high-level, structural guidance, while later steps benefit from detailed, appearance-level

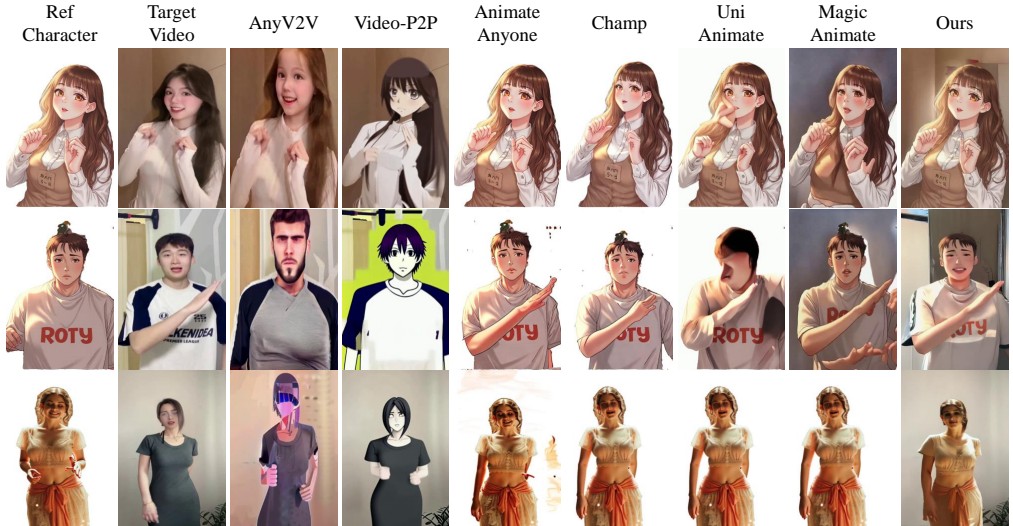

| Ref Character | Target Video | AnyV2V | Video-P2P | Animate Anyone | Champ | Uni Animate | Magic Animate | Ours |

Figure 4: Comparison of video character transfer. Note that our backgrounds are better preserved.

control. Then the denoising loss can be defined as follows:

$$\mathcal{L}(\theta) = \begin{cases} \|\epsilon - \hat{\epsilon}_\theta\left(z_t, \tau_{crs}, \mathcal{U}, t\right)\|_2^2, & t \in [t_c, T-1] \\ \|\epsilon - \hat{\epsilon}_\theta\left(z_t, \tau_{mid}, \mathcal{U}, t\right)\|_2^2, & t \in [t_f, t_c) \\ \|\epsilon - \hat{\epsilon}_\theta\left(z_t, \tau_{fine}, \mathcal{U}, t\right)\|_2^2, & t \in [0, t_f) \end{cases} \tag{3}$$

where $\tau_{crs}, \tau_{mid}, \tau_{fine}$ are coarse-grained, mid-grained and fine-grained text prompts, respectively. $t_c$ and $t_f$ are the end points of the corresponding stages. $T$ is the total timestep of the diffusion process. In our paper, as the dataset provides detailed video descriptions, we utilize them as the fine-grained prompts. Then we employ an LLM Qwen-QWQ-32B [38] to summarize $\tau_{fine}$ to $\tau_{crs}, \tau_{mid}$. This decomposition strategy helps to align the complexity of text guidance with the focus of the generation phase, improving both semantic consistency and visual fidelity in the final video output.

## 4 Experiments

To better demonstrate the strengths of our approach, we conduct experiments to validate foreground transfer, background transfer, and optical flow transfer across videos. The implementation details can be referred to the appendix.

### 4.1 Animal-centric Dataset

To promote the broader applicability of video editing models, we introduced a new dataset OpenAnimal, which focused on single-animal video sequences across a wide range of species and diverse motion patterns. While following a similar structure to human-centric datasets (e.g., TikTok, UBC), OpenAnimal is specifically tailored for animal subjects with 10000 video clips.

### 4.2 Comparison and Analysis

As current state-of-the-art methods are primarily trained on human datasets, they often struggle to generalize to non-human scenarios, such as animal motion transfer or object-level editing in nature-based scenes. For fair comparison, we conducted qualitative and quantitative experiments on UBC[50] and TikTok[17] to evaluate the video character transfer performance. Besides, to further evaluate our model's ability to generalize to other concepts, we also demonstrate qualitative results of animal transfer, cloth transfer, background transfer, and motion transfer.

### 4.2.1 Video Character Transfer

To evaluate the effectiveness of our method in video character transfer, we compare it with state-of-the-art character animation and video editing baselines, including Animate Anyone [14], Champ [57],

Table 1: Quantitative comparison of video quality for video character transfer.

| | TikTok | | | | | UBC | | | | |
|---|---|---|---|---|---|---|---|---|---|---|
| | LPIPS↓ | PSNR↑ | SSIM↑ | FID↓ | FVD↓ | LPIPS↓ | PSNR↑ | SSIM↑ | FID↓ | FVD↓ |
| MRAA | 0.513 | 25.31 | 0.425 | 134.23 | 634 | 0.589 | 23.32 | 0.537 | 89.13 | 438 |
| DisCo | 0.674 | 21.56 | 0.532 | 112.24 | 571 | 0.467 | 23.45 | 0.412 | 94.15 | 483 |
| MagicAnimate | 0.259 | 24.56 | 0.657 | 89.23 | 428 | 0.143 | 27.08 | 0.742 | 48.19 | 391 |
| Animate Anyone | 0.198 | 25.89 | 0.713 | 79.18 | 398 | 0.132 | 26.54 | 0.798 | 44.18 | 378 |
| Champ | 0.241 | 26.57 | 0.787 | 67.14 | 401 | 0.159 | 27.03 | 0.811 | 41.02 | 324 |
| UniAnimate | 0.191 | 24.34 | 0.724 | 58.19 | 378 | 0.127 | 27.09 | 0.798 | 43.24 | 334 |
| Ours | 0.152 | 26.78 | 0.803 | 46.74 | 345 | 0.125 | 27.12 | 0.814 | 39.73 | 312 |

Table 2: Video consistency quality comparison. SubC: Subject Consistency; BkgC: Background Consistency; MoS: Motion Smoothness; AesQ: Aesthetic Quality; DyaD: Dynamic Degree.

| | TikTok | | | | | UBC | | | | |
|---|---|---|---|---|---|---|---|---|---|---|
| | SubC↑ | BkgC↑ | MoS↑ | AesQ↑ | DyaD↑ | SubC↑ | BkgC↑ | MoS↑ | AesQ↑ | DyaD↑ |
| Video-P2P | 0.842 | 0.853 | 0.782 | 0.443 | 0.710 | 0.813 | 0.867 | 0.734 | 0.497 | 0.371 |
| ControlVideo | 0.712 | 0.419 | 0.747 | 0.519 | 0.823 | 0.814 | 0.773 | 0.895 | 0.624 | 0.241 |
| RF-Editor | 0.911 | 0.873 | 0.581 | 0.423 | 0.765 | 0.825 | 0.648 | 0.789 | 0.434 | 0.627 |
| MotionClone | 0.784 | 0.865 | 0.891 | 0.651 | 0.830 | 0.924 | 0.943 | 0.871 | 0.613 | 0.679 |
| CogVideoX | 0.927 | 0.904 | 0.926 | 0.557 | 0.842 | 0.911 | 0.917 | 0.911 | 0.663 | 0.902 |
| Mofa-Video | 0.923 | 0.917 | 0.962 | 0.593 | 0.837 | 0.923 | 0.879 | 0.957 | 0.612 | 0.829 |
| Ours | 0.945 | 0.931 | 0.962 | 0.651 | 0.903 | 0.939 | 0.942 | 0.971 | 0.664 | 0.911 |

UniAnimate [43], AnyV2V [20], Video-P2P [23], *etc.* . Qualitative results are presented in Figure 4. Given an input video, these existing approaches typically modify or replace objects based on textual or image-based prompts. As illustrated in Figure 4, the text-guided transfer methods fail to transfer the reference image to the video, while the image-guided baselines struggle to preserve the structural features or maintain the appearance of the backgrounds. In contrast, our approach leverages reference image guidance and incorporates a progressive spatial and timestep decomposition, which aligns more naturally with real-world editing workflows. Our approach can achieve better preservation of the reference appearance, higher visual quality of the video, and greater inter-frame consistency than all baselines. These improvements are attributed to our novel decomposition strategy, which enables fine-grained control and more structured video generation.

Besides, we perform further quantitative evaluation from two perspectives: video quality and temporal consistency. As shown in Table 1 and Table 2, our method achieves superior performance across multiple metrics, including FID, LPIPS, and subject consistency, aesthetic quality, *etc.* from vbench[16], significantly outperforming the compared baselines. This demonstrates the robustness and generalization ability of our approach in complex video character transfer tasks.

### 4.2.2 Adaptation to Various Video Concept Transfer Tasks.

Our proposed decomposition-based modeling and random masking pre-training mechanism enable our framework to effectively isolate and manipulate individual visual factors within a video, which can generalize beyond conventional foreground transfer. In this section, we provide more diverse qualitative results of a wide range of video concept transfer tasks, including motion transfer, background transfer, animal transfer, and regional foreground transfer, such as clothing replacement. These results demonstrate the flexibility and adaptability of our approach in handling diverse transformations, which are typically challenging for conventional video editing methods.

**Motion transfer.** Our framework supports motion transfer by applying the motion dynamics extracted from a driving video onto a reference image, similar to pioneer works[37]. To evaluate the effectiveness of our method on this task, we conduct comparisons with state-of-the-art motion-guided video generation approaches, including AnyV2V [20], MotionClone [22], MotionI2V [37], AnyV2V [20] and Mofa-Video [31]. The results are shown in Figure 5. As illustrated, MotionI2V struggles with maintaining the video consistency as time progresses, often introducing noticeable artifacts or drift in later frames. AnyV2V fails to preserve the appearance of the reference image. MotionClone primarily relies on text guidance for controlling video appearance, which limits its ability to precisely align the generated content with the reference image. Mofa-Video may generate blurry results. In contrast, our method produces videos that not only maintain high visual fidelity but also exhibit coherent and realistic motion consistent with the driving video. This advantage stems from our explicit spatial decomposition and motion-conditioned generation design, which enables better disentanglement and control over appearance and temporal dynamics.

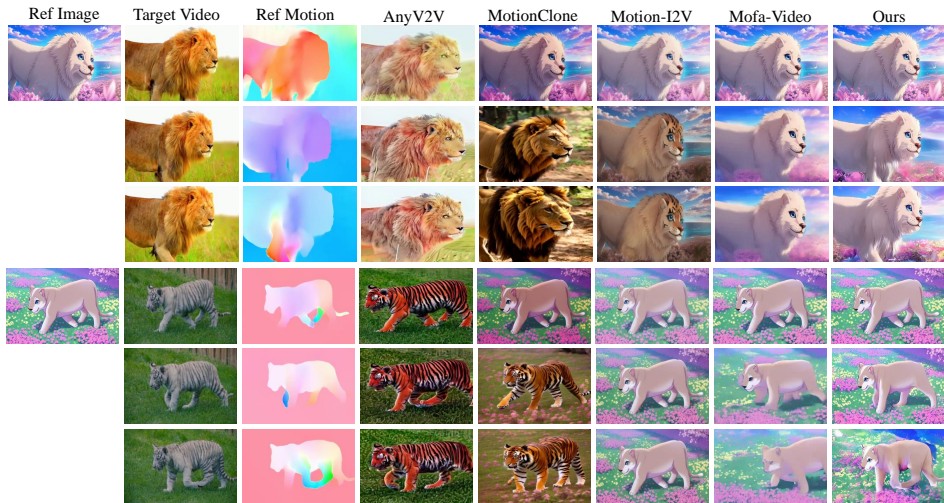

Figure 5: Comparison of video motion transfer.

Table 3: Ablation study.

| | Visual Quality | | | | | Video Smoothness | |
|---|---|---|---|---|---|---|---|
| | LPIPS ↓ | PSNR ↑ | SSIM ↑ | FID ↓ | FVD ↓ | SubC ↑ | BkgC ↑ |
| Early Motion Injection | 0.234 | 25.16 | 0.743 | 56.79 | 374 | 0.871 | 0.736 |
| w/o Flow Noise Injection | 0.293 | 24.31 | 0.659 | 69.31 | 391 | 0.915 | 0.893 |
| w/o Self-Supervised Pre-training | 0.498 | 19.87 | 0.546 | 101.34 | 487 | 0.735 | 0.892 |
| w/o dual-stream decomposition | 0.341 | 23.12 | 0.694 | 78.13 | 423 | 0.894 | 0.639 |
| w/o CoP Timestep Decomposition | 0.192 | 25.87 | 0.712 | 51.32 | 357 | 0.881 | 0.924 |
| Full Model | **0.152** | **26.78** | **0.803** | **46.74** | **345** | **0.945** | **0.931** |

**Foreground transfer.** Our framework enables flexible foreground transfer, including part-level object replacement, such as garment transfer. It poses significant challenges due to the need to selectively modify localized visual regions while preserving the subject's identity and overall video harmonization. This is particularly important in fashion, virtual try-on, and personalization applications. As shown in Figure 7, our model successfully replaces the clothing items in the target videos, with the referenced garments, without disrupting the consistency of facial features, body motion, or background. These results highlight the strength of our decomposition framework in enabling precise and high-quality object-level modifications within complex video generation tasks.

**Background transfer.** In most video editing approaches, modifications are typically limited to the main subject or local elements. In contrast, our method supports background replacement by leveraging the spatial disentanglement. As illustrated in Figure 7, our framework enables users to seamlessly replace the entire background of a video. Our approach ensures that background changes—such as scene transitions from indoor to outdoor—are consistent across all frames, while the foreground subject remains temporally coherent and unaffected in terms of identity and motion.

### 4.3 Ablation Study.

To verify the effectiveness of the components in our video generation pipeline, we designed several sets of ablation experiments on TikTok[17]. Specifically, we evaluate the following configurations: (1) Early Motion Injection: Instead of injecting motion in the intermediate network layers, we input motion information alongside the foreground and background features in the early layers, and apply attention-based fusion within the denoising branch. (2) Feature Fusion (w/o dual-stream decomposition): We directly concatenate foreground and background features and feed them into the denoising module. (3) Without Self-Supervised Pre-training. (4) Without CoP Timestep Decomposition. (5) Without Flow Noise Injection. The ablation study results are shown in Table 3. From the first two rows, we can see the superiority of the design of our motion branch, where the added noise improves the generalization ability and the injection strategy provides more accurate guidance. The third row emphasizes that the self-supervised training is critical for enhancing the

decomposed representation learning. The last two rows illustrate that our modeling of the spatial and timestep decomposition both are essential for flexible and high-quality video concept manipulation.

## 5 Conclusion

In this work, we present a novel framework for controllable video concept transfer by introducing a progressive spatial and timestep decomposition modeling strategy. Unlike existing methods that treat the video as a holistic entity, our approach explicitly disentangles the video into foreground, background, and motion components, and further decomposes the denoising process into hierarchical stages via chain-of-prompt guidance. This design allows for more fine-grained control over video synthesis, enabling diverse video concept transfer tasks, such as character transfer, motion transfer, background transfer, and garment-level editing. We also incorporate a self-supervised pretraining scheme based on random masking to support robust learning under no additional supervision, which effectively bootstraps spatial disentanglement using large-scale unlabeled video data. Besides, we curate a new dataset featuring animal subjects to advance research in video generation. Extensive experiments on different transfer tasks demonstrate that our method significantly outperforms state-of-the-art baselines in terms of visual quality, consistency, and controllability.

## Acknowledgments and Disclosure of Funding

Weiwei Xu is partially supported by NSFC (No. 92570206, No.62421003) . Rong Zhang is partially supported by NSFC (No. 62302449) and Zhejiang Provincial Natural Science Foundation of China (No. LQ23F020009). Zhiyuan Ma is supported by the National Natural Science Foundation of China (No. 62406161), China Postdoctoral Science Foundation (No. 2023M741950), and the Postdoctoral Fellowship Program of CPSF (No. GZB20230347). This paper is partially supported by the Yongjiang Innovation Project No. 2025Z062.

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

## A Video Diffusion Model Preliminary

Video diffusion models generalize the concept of image diffusion probabilistic models [13] to the temporal domain. Formally, let $z_0 \in \mathbb{V}^{f \times h \times w \times c}$ represent a video latent variable, where $f$ denotes the number of frames, with $h \times w$ dimension, $c$ channels. The forward diffusion process is defined as a Markov chain that gradually corrupts the original video into Gaussian noise:

$$z_t = \sqrt{\bar{\alpha}_t} z_{t-1} + \sqrt{1 - \bar{\alpha}_t} \epsilon, \quad \epsilon \sim \mathcal{N}(0, \mathbf{I}), \tag{4}$$

where $t \in \{1, \ldots, T\}$ indexes the diffusion timestep, $\bar{\alpha}_t$ controls the noise intensity, and $\epsilon$ represents standard Gaussian noise. In the reverse process, a denoising network is used to estimate the noise from $z_{t-1}$ to $z_t$, typically called a Diffusion Model $\theta$. The training objective minimizes the following loss function:

$$\mathcal{L}(\theta) = \mathbb{E}_{z_0, \epsilon, \mathcal{C}, t} \left[ \| \epsilon - \hat{\epsilon}_\theta(z_t, \mathcal{C}, t) \|_2^2 \right], \tag{5}$$

where $\mathcal{C}$ represents conditioning information such as text prompts or reference images.

## B Implementation Details

In this paper, we focus on image-guided video concept transfer, where the foregrounds, backgrounds, or motion dynamics can be manipulated. We perform experiments including character, animal, object, background and motion transfer to evaluate our approach. The weights of the model are initialized from a pretrained CogVideoX-5B [48]. The experiments are conducted on a server equipped with $8\times$ NVIDIA Tesla H100 80G GPUs. In our paper, we generate 49-frame videos at the resolution of $720 \times 480$.

Our training pipeline consists of two stages: a large-scale self-supervised pretraining phase and a supervised fine-tuning phase. In the pre-training phase, we trained our model on the OpenVid dataset[30], a large-scale collection of approximately 12 million videos covering a wide range of categories including humans, animals, vehicles, and diverse backgrounds. The training used a learning rate of $1 \times 10^{-4}$ and ran for $200,000$ iterations. The OpenVid dataset comprises approximately 12 million samples.

In the supervised training phase, we adapt the pretrained model to specific video concept transfer scenarios. For the human-centric editing task, we fine-tune the model on a combination of the TikTok [17] and UBC Fashion [50] datasets, which include rich annotations for character-level transfer tasks. The model is finetuned for $10,000$ iterations with a learning rate of $5 \times 10^{-5}$. For the animal-centric task, we performed an additional $10,000$ fine-tuning iterations on OpenAnimal using the same learning rate.

## C More Details about Self-Supervised Pretraining

In practice, obtaining large amounts of high-quality annotated data remains challenging. Although efficient segmentation tools like SAM2[35] are available, they still require extensive manual interactions, such as point prompts, bounding boxes, or object-specific filtering. To address this limitation and reduce the reliance on manual annotations, we introduce a self-supervised pretraining approach, the detailed algorithmic pipeline of which is outlined below Algorithm 1 and Figure 6.

## D More Details about CoP Guidance

The detailed algorithmic pipeline of our Chain-of-Prompt (CoP) guidance is demonstrated in Algorithm 2.

To assess the effectiveness of CoP, we conducted ablation experiments comparing it with static prompts, including: (i) All Coarse Prompt: using a coarse prompt across all timesteps, and (ii) All Detailed Prompt: using a detailed prompt across all timesteps. The results demonstrate that the all-coarse setting tends to produce results lacking fine-grained appearance fidelity and exhibiting texture degradation, while the all-detailed setting often introduces misguidance during early denoising steps, leading to distorted structures. In contrast, our CoP strategy, which progressively schedules coarse-to-fine prompts, strikes a balance between structural stability and appearance precision, achieving better performance. The quantitative evaluation can be referred to in Table 4.

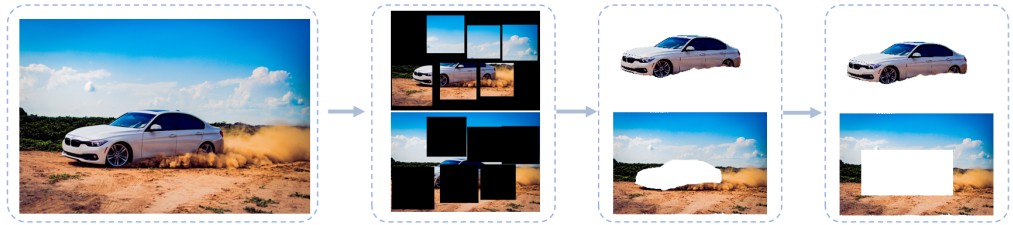

Figure 6: Self-supervised Pretraining.

# E   More Results

Our framework enables flexible foreground and background transfer, including part-level object replacement, such as garment transfer. The results of different transfer tasks are shown in Figure 10, Figure 11, Figure 12, Figure 7. Our curated animal-centric dataset OpenAnimal is demonstrated in Figure 13.

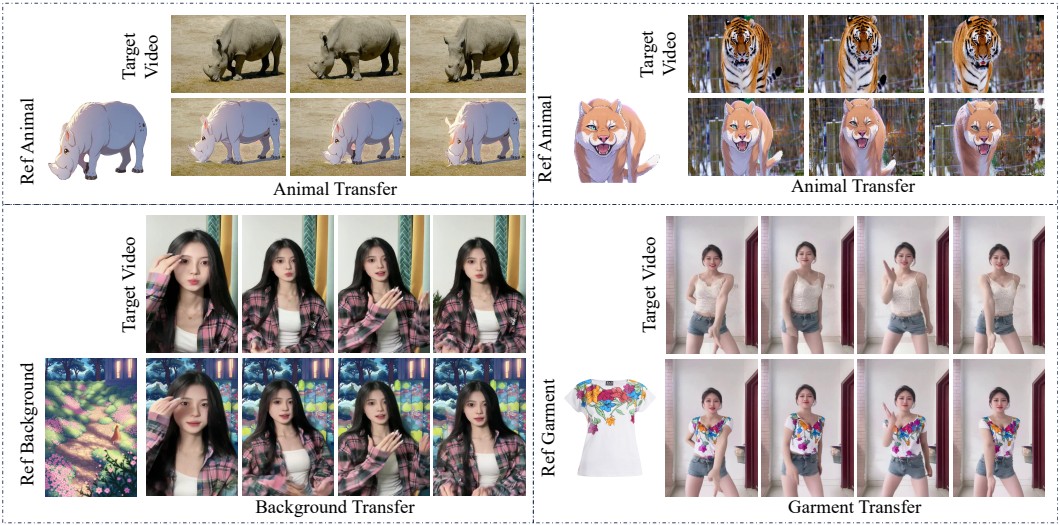

Figure 7: More UniTransfer results of different transfer tasks.

# F   Limitation

Although our model can achieve subject transfer and background replacement in videos, there are still cases where the subject and background appear with artifacts. This issue might be due to current segmentation models cannot fully separate foreground and background elements, leading to imperfect composite results in the final output. In the future, we plan to address this by leveraging large-scale models for enhanced video scene understanding, further improving the quality of generated videos.

For scenes involving multiple moving entities, we can fuse their individual optical flows into a composite representation and sequentially condition the network on each object through multiple inference passes. For interacting subjects, our method remains effective when the occlusion or overlap between them is relatively small, as each subject's region can still be identified and modeled with reasonable independence. However, cases with large overlapping areas or complex physical interactions pose challenges to our models due to ambiguity in motion attribution and visual entanglement. Joint modeling of interacting subjects is an interesting direction for future work.

**Algorithm 1:** The pipeline of our self-supervised pretraining.

**Input:** $image$: input image
$coverage$: target coverage ratio (default: 0.5)
$min\_block\_size$: minimum block size (default: 320)
$max\_block\_size$: maximum block size (default: 640)
**Output:** $foreground$: processed image with coverage
$background$: inverse masked image
**Function** $random\_white\_blocks(image, coverage, min\_block\_size, max\_block\_size)$

   **if** *image is None* **then**
      | raise ValueError("Input image is empty");

   **if** *image is grayscale* **then**
      | $result \leftarrow$ convert $image$ to BGR color space;
   **else**
      | $result \leftarrow$ copy of $image$;

   $(h, w) \leftarrow$ height and width of $result$;
   $total\_area \leftarrow h \times w$;
   $covered\_area \leftarrow 0$;
   $target\_area \leftarrow total\_area \times coverage$;
   $mask \leftarrow$ zero matrix of size $(h, w)$;
   $result\_2 \leftarrow$ gray matrix (127.5) with same size as $result$;
   $max\_iter \leftarrow 500$;
   **while** $covered\_area < target\_area$ ***and*** $max\_iter > 0$ **do**
      | $max\_iter \leftarrow max\_iter - 1$;
      | $block\_size \leftarrow$ random integer between $min\_block\_size$ and $max\_block\_size$;
      | $x \leftarrow$ random integer between 0 and $w - block\_size$;
      | $y \leftarrow$ random integer between 0 and $h - block\_size$;
      | **if** $mask[y : y + block\_size, x : x + block\_size]$ *contains any* 1 **then**
         | continue;
               $\triangleright$ Special overlapping effect $result\_2[y : y + block\_size - 5, x : x + block\_size - 5] \leftarrow result[y : y + block\_size - 5, x : x + block\_size - 5]$;
      | $result[y : y + block\_size + 5, x : x + block\_size + 5] \leftarrow [127.5, 127.5, 127.5]$;
      | $mask[y : y + block\_size, x : x + block\_size] \leftarrow 1$;
      | $covered\_area \leftarrow covered\_area + block\_size \times block\_size$;
      | **if** $covered\_area > 1.1 \times target\_area$ **then**
         | break;

   **return** $foreground$, $background$;

Table 4: CoP mechanism with different level prompts.

| | Visual Quality | | | | | Video Smoothness | |
|---|---|---|---|---|---|---|---|
| | LPIPS ↓ | PSNR ↑ | SSIM ↑ | FID ↓ | FVD ↓ | SubC ↑ | BkgC ↑ |
| All Coarse Prompt | 0.174 | 23.15 | 0.701 | 72.46 | 433 | 0.714 | 0.865 |
| All Detailed Prompt | 0.165 | 24.62 | 0.734 | 49.26 | 367 | 0.832 | 0.901 |
| Full Model | **0.152** | **26.78** | **0.803** | **46.74** | **345** | **0.945** | **0.931** |

## G Social Impact

Our research decouples video generation into foreground, background, and their corresponding motion. This technology will enhance the efficiency of video production, drive innovation in industries such as film and gaming, and enable more immersive entertainment and educational experiences. However, as this technology becomes more widespread, society may face ethical and legal challenges, including concerns over the authenticity of video content, characters, and backgrounds. Therefore, establishing appropriate regulatory frameworks to ensure responsible use of this technology is a critical task that demands our attention.

**Algorithm 2:** Chain-of-Prompt-Guided Video Denoising

**Input:** *initial_noisy_video*: Initial noisy video
*base_prompt*: Original text description
*total_steps*: Total denoising steps (default: 50)
$T1, T2$: Stage transition steps
**Output:** *generated_video*: Final generated video
**Function** *hierarchical_denoising(initial_noisy_video, base_prompt, total_steps = 50)*
    // Phase partitioning (all steps) $T1 \leftarrow 35$ ;                  `// First phase`
    $T2 \leftarrow 15$ ;                               `// Second phase`
    // Generate hierarchical prompts using LLM
    *prompts* ← QwQ32B_GenerateHierarchicalPrompts(*base_prompt*) ;
    `// Returns: {'stage1':coarse, 'stage2':detailed, 'stage3':fine}`
    *current_video* ← *initial_noisy_video*;
    **for** *step* ← *total_steps* **to** 1 **do**
        **if** $step \geq T1$ **then**
            *guidance_prompt* ← *prompts*['*stage1*'] ;       `// Coarse prompt`
            *guidance_weight* ← 1.5 ;           `// Strong guidance`
        **else if** $step \geq T2$ **then**
            *guidance_prompt* ← *prompts*['*stage2*'] ;       `// Detailed prompt`
            *guidance_weight* ← 2.0;
        **else**
            *guidance_prompt* ← *prompts*['*stage3*'] ;       `// Fine prompt`
            *guidance_weight* ← 1.0;
        // Execute denoising step *current_video* ←
          DenoiseStep(*current_video*, *guidance_prompt*, *guidance_weight*, *step*);
    **return** *current_video*;

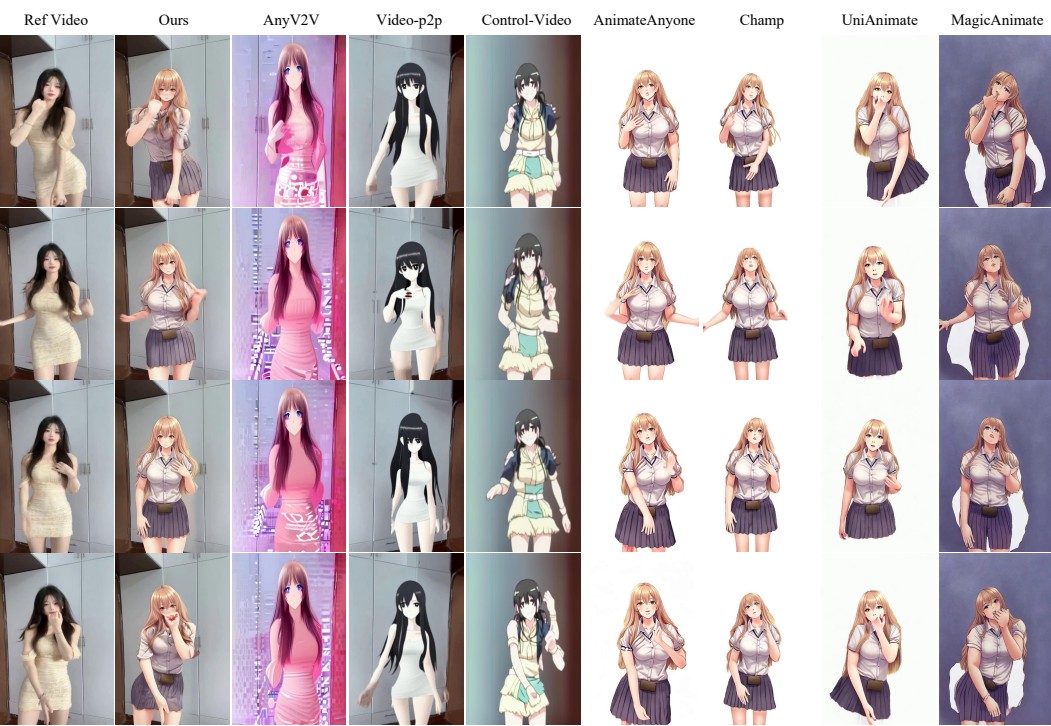

Figure 8: More results compared with other methods.

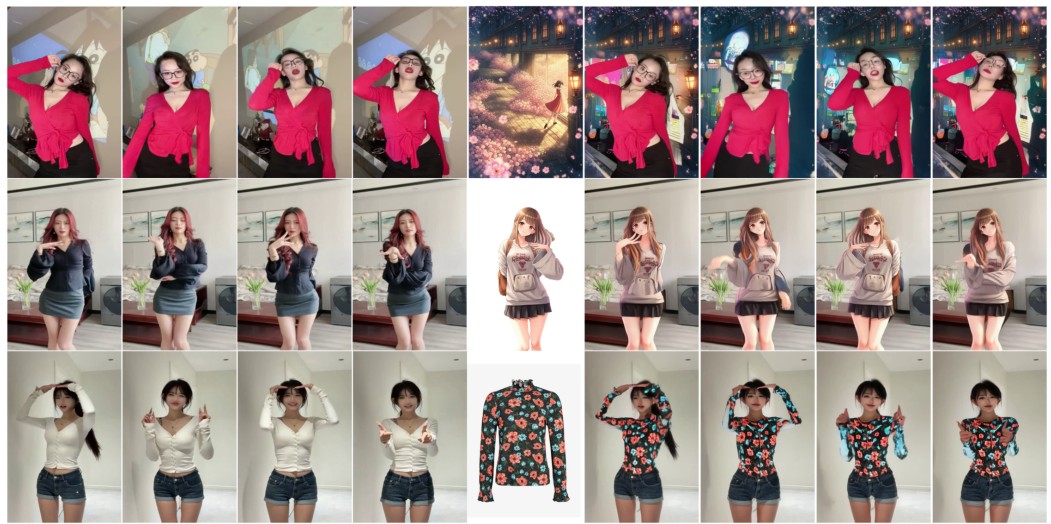

Figure 9: More Visual Results (Left is reference video, middle is reference image, right is output).

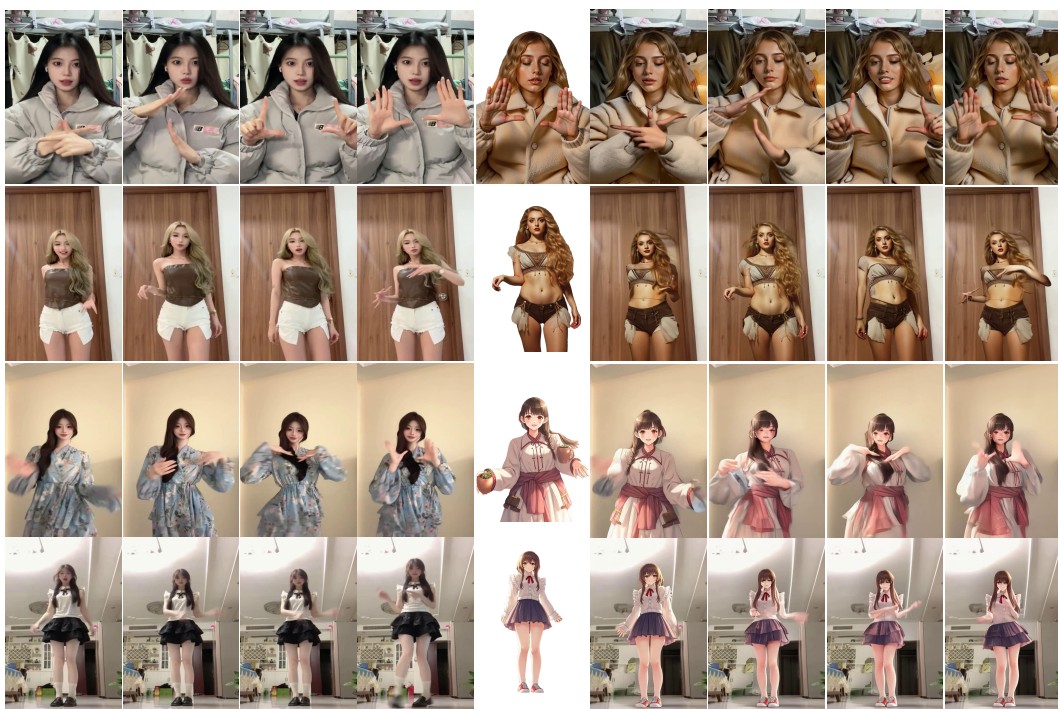

Figure 10: More Visual Results.(Left is reference video, middle is reference image, right is output)

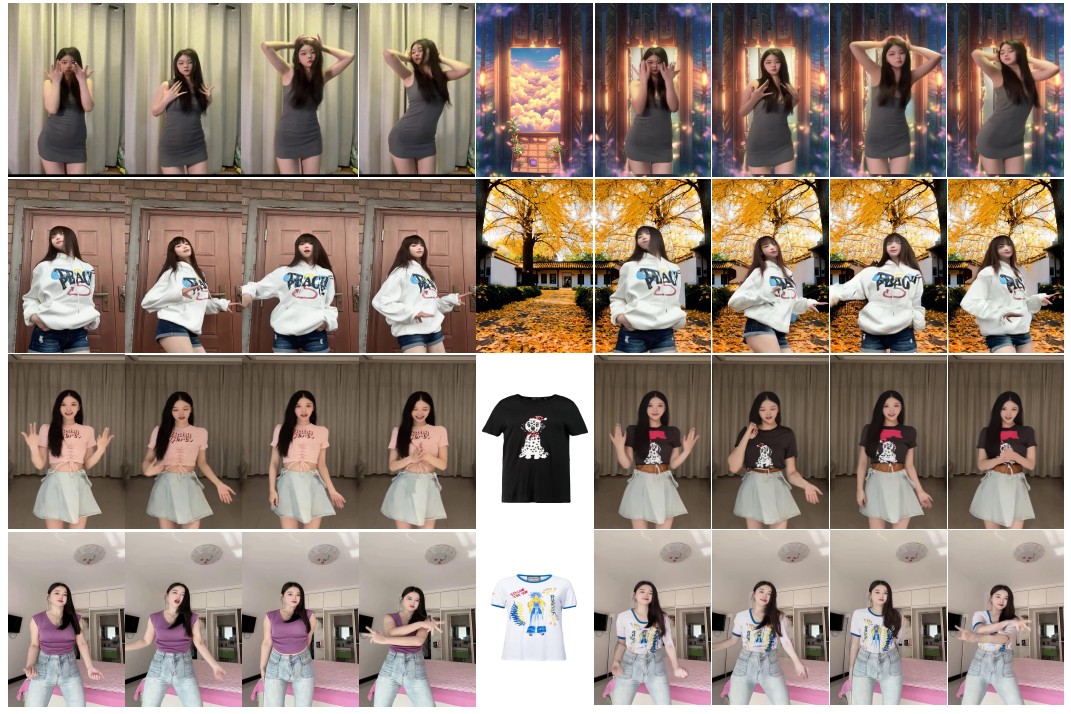

Figure 11: More Visual Results.(Left is reference video, middle is reference image, right is output)

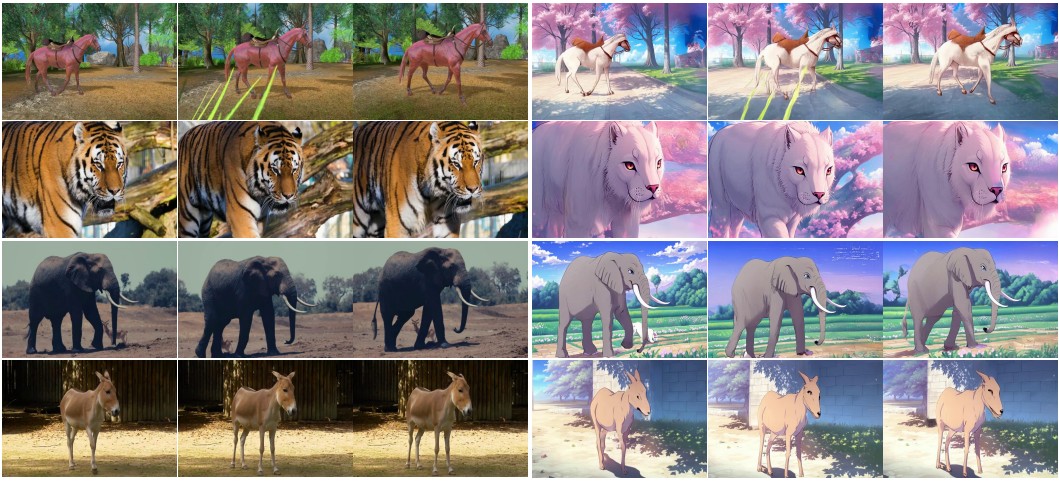

Figure 12: Animal Motion Transfer.(Left is reference video, right is output)

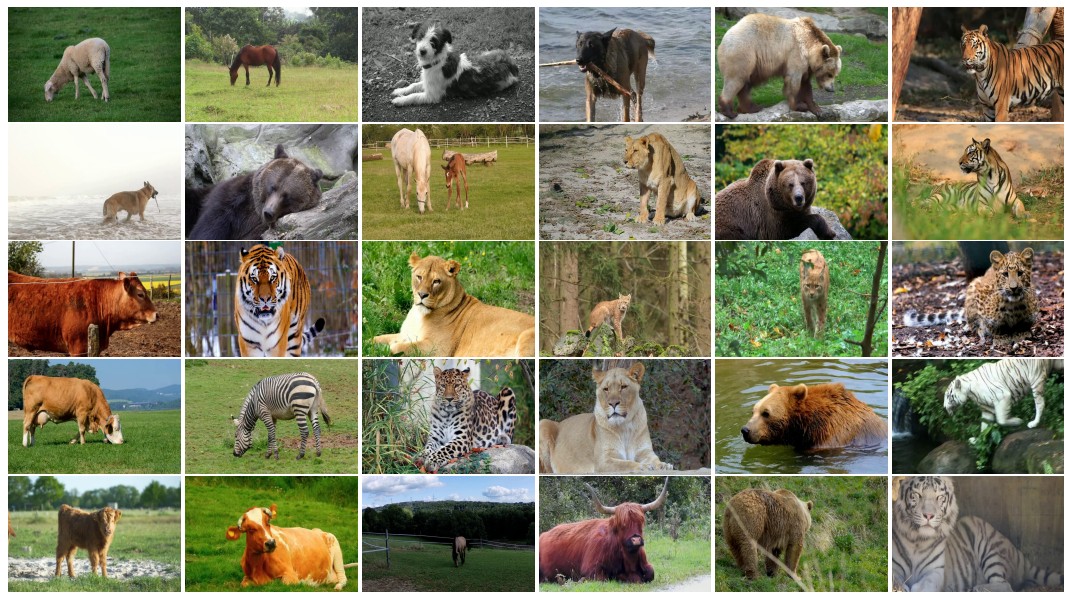

Figure 13: OpenAnimals Datasets.

