# OpenReview forum: "UniTransfer: Video Concept Transfer via Progressive Spatio-Temporal Decomposition"
_NeurIPS.cc/2025/Conference — NeurIPS 2025 poster_

### Official Review · Reviewer_Cz3v · 2025-06-29

**Clarity:** 3
**Significance:** 3
**Originality:** 3
**Rating:** 5
**Confidence:** 3

**Summary:**

This paper introduces UniTransfer, a video generation framework that supports controllable concept transfer (e.g., character, motion, garment, and background) using image-based references. Extensive experiments across TikTok, UBC, and a newly released OpenAnimal dataset show performance gains over baseline methods.

**Questions:**

1. What evidence supports that CoP improves generation beyond standard prompting?

Please compare with static prompts, or learnable embeddings. How robust are the LLM prompts under rephrasing or minor perturbations?

2. How deterministic is CoP?

Given the stochastic nature of LLMs, how do different CoP generations affect the final result? Do prompt variations lead to style shifts or artifacts?

**Ethical Concerns:**

["NO or VERY MINOR ethics concerns only"]

**Final Justification:**

The authors have addressed my concerns by providing a complete rebuttal. I would be happy to raise my score.

**Limitations:**

Yes

**Quality:**

3

**Strengths And Weaknesses:**

Strengths
1. Clear motivation and modular design: Tackles a real problem in video generation—lack of flexible and fine-grained control.

2. Methodologically comprehensive: Combines spatial disentanglement, motion conditioning, and progressive textual control into a unified pipeline.

3. Solid empirical performance: Competitive results across several tasks (character transfer, background replacement, garment/motion transfer).

4. Dataset contribution: OpenAnimal provides non-human evaluation data and enhances the paper’s experimental diversity.


Weaknesses:

1. The CoP mechanism is conceptually interesting but remains weakly justified. No baseline comparison (e.g., single prompt or learned embeddings), nor any analysis of prompt quality, semantic drift, or hallucination is provided.
2. The use of LLMs for generating stage-specific prompts introduces non-determinism, inference overhead, and variability, which are not quantified or discussed.

---

> ### Author Rebuttal · Authors · 2025-07-31
>
> (1) The CoP mechanism is conceptually interesting but remains weakly justified. No baseline comparison (e.g., single prompt or learned embeddings), nor any analysis of prompt quality, semantic drift, or hallucination is provided. Please compare with static prompts, or learnable embeddings. How robust are the LLM prompts under rephrasing or minor perturbations?
>
> **Answer:**
> Thank you for your insightful suggestions regarding the CoP mechanism.
>
> - **Comparison with Static Prompts**
>
>   To assess the effectiveness of CoP, we conducted ablation experiments comparing it with static prompts, including:
>   (i) using coarse prompt across all timesteps, and (ii) using detailed prompt across all timesteps. The results demonstrate that the all-coarse setting tends to produce outputs that lack fine-grained appearance fidelity and exhibit texture degradation, while the all-detailed setting often introduces misguidance during early denoising steps, leading to distorted structure.
>   In contrast, our CoP strategy, which progressively schedules coarse-to-fine prompts, balances structural stability with appearance precision and achieves better performance both qualitatively and quantitatively.
>   The quantitative evaluation can be referred to in the following table.
>
> - **Robustness to Prompt Rephrasing**
>
>   We further evaluated the robustness of the LLM-generated prompts under rephrasing and textual perturbations. The results indicate that mid- and fine-level prompts are generally robust to rewording. In contrast, coarse-level prompts are relatively more sensitive to rephrasing as they are applied during the early timesteps and primarily guide the global structure. Rephrasing at this stage may lead to structural shifts or artifacts, since the deviations in the early steps tend to accumulate during the denoising process.
>
> **Cop mechanism On Different Methods.** SubC: Subject Consistency; BkgC: Background Consistency; MoS: Motion Smoothness; AesQ: Aesthetic Quality; DyaD: Dynamic Degree.
>
> |                                 |       |       | CogVideoX |       |       |      |       |       | MotionClone |       |       |
> | :-----------------------------: | ----- | :---: | --------- | ----- | ----- | ---- | ----- | ----- | ----------- | ----- | ----- |
> |                                 | SubC  | BkgC  | Mos       | AesQ  | DyaD  |      | SubC  | BkgC  | MoS         | AesQ  | DyaD  |
> |        All Coarse Prompt        | 0.853 | 0.892 | 0.765     | 0.512 | 0.734 |      | 0.712 | 0.823 | 0.657       | 0.425 | 0.689 |
> |       All detailed Prompt       | 0.911 | 0.912 | 0.662     | 0.511 | 0.501 |      | 0.789 | 0.871 | 0.881       | 0.662 | 0.842 |
> | with CoP Timestep Decomposition | **0.917** | **0.921** | **0.893**     | **0.559** | **0.893** |      | **0.836** | **0.889** | **0.912**       | **0.683** | **0.864** |
>
> ​																				**CoP mechanism with different level prompts.**
>
> |                     |       |       | Visual Quality |       |      |      | Video Smoothness |       |
> | :-----------------: | ----- | ----- | :------------: | ----- | ---- | ---- | :--------------: | ----- |
> |                     | LPIPS | PSNR  |      SSIM      | FID   | FVD  |      |       SubC       | BkgC  |
> |  All Coarse Prompt  | 0.174 | 23.15 |     0.701      | 72.46 | 433  |      |      0.714       | 0.865 |
> | All Detailed Prompt | 0.165 | 24.62 |     0.734      | 49.26 | 367  |      |      0.832       | 0.901 |
> |     Ours with CoP Timestep Decomposition      | **0.152** | **26.78** |     **0.803**      | **46.74** | **345**  |      |      **0.945**       | **0.931** |
>
> ---
>
> (2) The use of LLMs for generating stage-specific prompts introduces non-determinism, inference overhead, and variability, which are not quantified or discussed. How deterministic is CoP? Given the stochastic nature of LLMs, how do different CoP generations affect the final result? Do prompt variations lead to style shifts or artifacts?
>
> **Answer:**
> Thank you for raising important concerns regarding the use of LLMs in the Chain-of-Prompts (CoP) mechanism.
>
> - **Inference Overhead**
>
>   While it is true that incorporating LLMs introduces additional computation, we found the overall inference overhead to be minimal. For instance, generating a 49-frame video at a resolution of $720\times480$ using an Nvidia-A100 GPU takes approximately 215 seconds, whereas LLM prompt generation only accounts for around 5 seconds, i.e., about 2\% of the total runtime. This small cost is acceptable given its benefits.
>
> - **CoP Certainty**
>
>   To assess the certainty and impact of LLM stochasticity on CoP, experiments are conducted in which multiple CoP prompt sequences were generated using different random seeds for the same input video and reference image. But the resulting videos were visually and structurally consistent, and we did not observe significant style shifts or artifacts attributable to LLM-induced prompt changes. This supports the conclusion that CoP provides a robust and practically deterministic interface for guiding generation.
>
> A likely reason for this robustness may lie in the sensitivity to different prompt stages. As discussed earlier, mid- and fine-level prompts are generally robust to rewording, while the coarse-level prompts (applied in early denoising timesteps) have a greater influence on the results. But coarse-level prompts are typically short and simple (usually <10 words), making them less susceptible to LLM variability.

---

> > ### Comment · Reviewer_Cz3v · 2025-08-03
> >
> > Thanks for the rebuttal. Most of concerns have been resolved.

---

> > > ### Author Response · Authors · 2025-08-05
> > >
> > > Thank you for your timely feedback. We're glad to hear that most of your concerns have been resolved. If there are any remaining questions or concerns, please feel free to reach out to us.

---

### Official Review · Reviewer_djA4 · 2025-06-30

**Clarity:** 3
**Significance:** 3
**Originality:** 3
**Rating:** 5
**Confidence:** 4

**Summary:**

This paper introduces UniTransfer, an image-guided video generation framework that achieves controllable video concept transfer by progressively decomposing both the spatial content of a video by separating each clip into foreground, background and motion flow; and diffusion denoising timeline by applying a chain-of-prompt strategy. To reduce annotation effort, the model first trained on self-supervised random mask scheme , then fine-tuned with limited masks and optical flow. They also release a 10k-clip dataset of single-animal videos.

**Questions:**

- Please specify the timestep boundaries (T1, T2) and the prompt lengths produced by the LLM for each stage. How sensitive are results to (i) the choice of those boundaries and (ii) the quality or length of the LLM-generated coarse/mid prompts.
- Report end-to-end inference time, peak GPU memory, and FLOPs for generating a 16-frame 512×512 video.
- The current pipeline assumes one foreground mask. Can UniTransfer handle scenes with two interacting subjects (e.g. two people, or one animal-one person, or multiple subjects ) or must it be run sequentially per subject.
- Only qualitative results are shown for these tasks. Could you report objective metrics versus any baselines where applicable?
- Include a brief section (main or appendix) with representative failure modes (e.g. extreme perspective mismatch, inaccurate optical flow, or misleading LLM summaries) and describe potential remedies.

**Ethical Concerns:**

["NO or VERY MINOR ethics concerns only"]

**Final Justification:**

I thank the authors for their detailed rebuttal. After careful reflection on their response and the nature of the paper's contribution, I have decided to increase my score to 5.
My initial primary concern was the lack of quantitative evaluation for tasks like garment and background transfer. While this remains a valid point, I have reconsidered its weight relative to the paper's overall contribution. The core innovation of this work is the UniTransfer framework, a novel and technically sound architecture that demonstrates remarkable versatility across four distinct concept transfer tasks. The authors successfully show this flexibility qualitatively, which is the primary evidence for their main claim.

Furthermore, the rebuttal effectively addressed my other concerns regarding computational cost, model limitations, and failure modes, demonstrating a high degree of transparency and rigor. Given this, I find that penalizing the paper for not conforming to evaluation metrics from more specialized domains may be overly rigid. The paper's contribution to general-purpose, controllable video generation is significant. The strengths in its novel architecture and demonstrated versatility now, in my view, outweigh the limitations in its evaluation strategy. Therefore, I now lean more towards acceptance.

**Limitations:**

The paper lists a basic limitation (artifact‐prone foreground/background compositing) and notes potential misuse of synthetic videos, but some important points remain unaddressed:
- Computational demand and reproducibility:
- Single-foreground assumption.
- Performance depends on accurate optical flow and LLM prompt quality.

**Paper Formatting Concerns:**

Authors added an appendix between the references and the paper checklist in the main PDF. This might break the 9-page limit.

**Quality:**

3

**Strengths And Weaknesses:**

Strengths:
- The paper tackles with the long-standing challenge of flexible, high-fidelity video editing with a dual-decomposition strategy that is both technically sound and motivated.
- By splitting generation along spatial (foreground, background, motion) and temporal (coarse-to-fine denoising) axes, the authors deliver a unified framework that handles character, garment, background and motion transfer in a single model.
- The proposed dual-to-single-stream DiT architecture and Chain-of-Prompt guidance are rigorously described, backed by equations, ablations and intuitive figures, reflecting high quality and clarity.
- The self-supervised random-mask pre-training further reduces annotation cost and is validated to be essential, adding practical value beyond raw performance.

Weaknesses:
- The framework depends on a three-branch diffusion transformer, a large language model for prompt decomposition, and an external optical-flow estimator, which raises the bar for reproducibility and deployment, diminishing practical quality despite strong results.
- Some important scenarios remain unexplored: the method assumes a single foreground subject and largely static backgrounds, so its behaviour with multiple moving objects, heavy camera motion, or depth-layered scenes is unclear, limiting its claimed significance.
- Quantitative evaluation is thorough for character replacement but only qualitative for garment and background transfer, leaving a gap in empirical evidence for those tasks and slightly weakening overall clarity of validation.
- The paper also omits discussion of inference cost and LLM latency, and gives limited insight into failure modes of the Chain-of-Prompt mechanism.

---

> ### Author Rebuttal · Authors · 2025-07-31
>
> (1) The framework depends on a three-branch diffusion transformer, a large language model for prompt decomposition, and an external optical-flow estimator, which raises the bar for reproducibility and deployment, diminishing practical quality despite strong results.
>
> **Answer**:
> 	Thank you for your comments. To ensure reproducibility, the full codebase will be released to the community.
>
> ---
>
> (2) Report end-to-end inference time, peak GPU memory, and FLOPs for generating a 16-frame 512×512 video.
>
> **Answer**:
> 	In our paper, we generate 49-frame videos at the resolution of $720\times480$. For a 16-frame $512\times512$ video, the inference time is about 60 seconds, and the peak GPU memory is about 22G with 23.42 TFLOPS using Nvidia-A100 series GPU.
>
> ---
>
> (3) Some important scenarios remain unexplored: the method assumes a single foreground subject and largely static backgrounds, so its behaviour with multiple moving objects, heavy camera motion, or depth-layered scenes is unclear, limiting its claimed significance. Can UniTransfer handle scenes with two interacting subjects (e.g. two people, or one animal-one person, or multiple subjects ) or must it be run sequentially per subject.
>
> **Answer**:
> 	For scenes involving multiple moving entities, we can fuse their individual optical flows into a composite representation and sequentially condition the network on each object through multiple inference passes. For interacting subjects, our method remains effective when the occlusion or overlap between them is relatively small, as each subject’s region can still be identified and modeled with reasonable independence. However, cases with large overlapping areas or complex physical interactions pose challenges to our models due to ambiguity in motion attribution and visual entanglement.
> We acknowledge this as a current limitation and joint modeling of interacting subjects is an interesting direction for future work.
>
> ---
>
> (4) Quantitative evaluation is thorough for character replacement but only qualitative for garment and background transfer, leaving a gap in empirical evidence for those tasks and slightly weakening overall clarity of validation. Could you report objective metrics versus any baselines where applicable?
>
> **Answer**:
> ​		Our method is designed for the video concept transfer task in general scenarios. While we did not target garment or background transfer specifically, our method naturally generalizes to these tasks, as shown in the teaser of our paper. Direct quantitative comparison with prior methods is inherently unfair to both sides: those methods rely on small, narrow-domain datasets and incorporate task-specific structures and losses (e.g., structure preserving module and garment consistent losses), while our model is trained on large-scale, general-purpose data without such targeted structures and losses.
> As a result, our approach supports broader generalization, though it does not explicitly enforce fine-grained consistency. Adding task-specific components could improve metric scores on these subtasks, but would compromise generality. Exploring such lightweight plugin-style extensions without compromising the generality of the core model is a promising future direction.
>
> ---
>
> (5) The paper also omits discussion of inference cost and LLM latency, and gives limited insight into failure modes of the Chain-of-Prompt mechanism. Include a brief section (main or appendix) with representative failure modes (e.g. extreme perspective mismatch, inaccurate optical flow, or misleading LLM summaries) and describe potential remedies.
>
> **Answer**:
> 		While LLMs add some computation, the overhead is minimal, with prompt generation taking only 5 seconds out of 215 (∼2\%) for a 49-frame $720\times480$ video on an NVIDIA-A100 GPU, which is acceptable given the benefits.
> We will include a brief section on representative failure modes (F) and potential remedies (R), as following:
>
> - **F: Extreme Perspective or Pose Mismatch.** R: Adopt a perspective/pose alignment via generative transfer or image personalization methods as a pre-processing step.
>
> - **F: Inaccurate Optical Flow.** R: Introduce auxiliary guidance from image features, depth estimation, semantics, and multi-frame correlations to enhance motion estimation in ambiguous regions.
>
> - **F: Misleading LLM Summaries.** To evaluate the impact of LLM stochasticity on CoP, we generated multiple prompt sequences using different random seeds for the same input. The resulting videos remained visually and structurally consistent, with no noticeable style shifts or artifacts, indicating that CoP offers a robust and near-deterministic control interface. This robustness likely stems from prompt stage sensitivity: while coarse-level prompts influence results more, they are typically short and simple, thus less affected by LLM variability.
>   Under extreme conditions, a potential remedy is to introduce an independent LLM-based verification agent to assess the plausibility of generated LLM summaries.
>
> - **F: Interacting Objects with Significant Overlap.** R: Auxiliary pose and depth information can be introduced alongside optical flow to better capture the spatial occlusion relationships between multiple objects..
>
> ---
>
> (6) Please specify the timestep boundaries (T1, T2) and the prompt lengths produced by the LLM for each stage. How sensitive are results to (i) the choice of those boundaries and (ii) the quality or length of the LLM-generated coarse/mid prompts.
>
> **Answer**:
> - Since different tasks and denoising strategies may be better suited to different timestep boundaries, we adopt a generally applicable scheme resembling a three-way split as specified in Algorithm.2 of the supplementary material. To assess the sensitivity of the boundaries, we conduct ablation studies by varying either $T_1$ or $T_1$ individually. Experimental results reveal that  $T_1$ has a greater impact on the generated videos and is more sensitive than $T_2$, as coarse prompts are applied during the early timesteps and guide the overall generation process.
>   We will include this discussion and the corresponding results in the revision.
> - Our current multi-level prompts are generated with LLM based on semantic hierarchy. The reviewer’s suggestion to further control prompt length and quality is constructive, as it has the potential to enhance the analytical clarity and ensure consistency across tasks. In response, we plan to include comparative experiments with explicit constraints on prompt length and sentence structure (e.g., the number and arrangement of subject-verb-object and adjective) in the revision.
>
> ---
>
> (7) Limitation: The paper lists a basic limitation (artifact‐prone foreground/background compositing) and notes potential misuse of synthetic videos, but some important points remain unaddressed: Computational demand and reproducibility/ Single-foreground assumption/ Performance depends on accurate optical flow and LLM prompt quality.
>
> **Answer**:
> 	We will include detailed computational demand in the revised version and release the code to ensure reproducibility. In addition, we will include the discussion of the *single-foreground assumption* and *performance dependence*, to further explore the boundary of our method.

---

### Official Review · Reviewer_Lpcd · 2025-07-03

**Clarity:** 3
**Significance:** 3
**Originality:** 2
**Rating:** 4
**Confidence:** 3

**Summary:**

The paper proposes UniTransfer, a novel architecture for precise and controllable video concept transfer. It spatially decomposes videos into foreground, background, and motion flow, using a dual-to-single-stream DiT-based architecture for fine-grained control. It also employs a "Chain-of-Prompt" mechanism for timestep decomposition, leveraging LLMs to guide the denoising process progressively. The paper also introduces the OpenAnimal dataset.

**Questions:**

- Is it reasonable to understand 'Sec. 3.1.2 Self-supervised Pre-training via Random Masking' as finetuning the vdm to learn inpainting prior?
- Is the model trained from scratch or finetuned from a pretrained video diffusion model? If the model is trained from scratch, why couldn't the authors finetune from a pretrained video model?

**Ethical Concerns:**

["NO or VERY MINOR ethics concerns only"]

**Final Justification:**

The paper proposes an unified framework for various concept transfer tasks, in addition to a new benchmark dataset. I believe the contributions will be valued by the community. Moreover, the rebuttal answered all my remaining questions. Hence, I would like to keep my rating.

**Limitations:**

- The paper omits discussing a highly relevant video concept personalization paper "Dynamic Concepts Personalization from Single Videos". What are the strength/limitation compared to this paper?
- The paper omits discussing pioneering video motion transfer works, such as "MotionDirector: Motion Customization of Text-to-Video Diffusion Models", "Space-Time Diffusion Features for Zero-Shot Text-Driven Motion Transfer", and "VMC: Video Motion Customization using Temporal Attention Adaption for Text-to-Video Diffusion Models". Specifically, how does the method performs compared to MotionDirector?

**Quality:**

3

**Strengths And Weaknesses:**

[Strength]
- The paper proposes a single integrated model to perform various concept transfer tasks.
- The authors collect OpenAnimal dataset o facilitate the training and benchmarking of research community.

[Weakness]
- For character transfer or video motion transfer, the reference input images are quite spatially aligned with the target video (e.g., fig1 top left, fig 4, fig 5). Is it necessary to have spatially-aligned reference input image? How does UniTransfer perform given spatially-unaligned reference input image?
- Regarding timestep decomposition, can the authors discuss why just providing a long, detailed prompt that contains all information of high-level, structural guidance, detailed, appearance-level description for all timesteps does not work? In other works, can the authors provide comparison experiments against this approach, which is similar to LLM-based prompt enhancing practice of recent vdms?

---

> ### Author Rebuttal · Authors · 2025-07-31
>
> (1) For character transfer or video motion transfer, the reference input images are quite spatially aligned with the target video (e.g., fig1 top left, fig 4, fig 5). Is it necessary to have spatially-aligned reference input image? How does UniTransfer perform given spatially-unaligned reference input image?
>
> **Answer：**
> 	For character or video motion transfer, our method can handle spatial misalignment between reference images and target videos (see videos on our anonymous website for examples). This is achieved by detecting the minimum bounding boxes of the main subjects in both the reference image and the first frame, then our model adjusts these boxes to align them through affine transformations.But significant misalignment truly poses a challenge for our method and the current research community.In such cases, a potential solution is to first perform pose alignment via pose transfer or image personalization methods as a pre-processing step.
>
> ---
>
> (2) Regarding timestep decomposition, can the authors discuss why just providing a long, detailed prompt that contains all information of high-level, structural guidance, detailed, appearance-level description for all timesteps does not work? In other works, can the authors provide comparison experiments against this approach, which is similar to LLM-based prompt enhancing practice of recent vdms?
>
> **Answer**：
> Our timestep decomposition strategy is designed in terms of two aspects:
>
> - **Task Characteristic.** Our study specifically targets the concept transfer task, where the goal is to seamlessly merge two distinct images guided by different prompts. In this context, providing fine-grained textual prompts for either the foreground object or the background image may introduce misguidance to the model and hinder the effective feature fusion.
>   To overcome this limitation, we adopt a timestep decomposition strategy.
> - **Diffusion Model Characteristic.** The strategy is grounded in empirical and theoretical insights from prior diffusion literature *Prospect*, which observes that early timesteps in the denoising process are responsible for generating structural layouts, while later timesteps refine details such as texture and appearance. Based on this, we hypothesize that providing coarse, high-level prompts in early stages helps avoid conflicting guidance, thus stabilizing the structure generation process.
>   In contrast, providing a fully detailed prompt at every timestep can lead to confusion or prompt oversaturation, where certain aspects of the prompt may be neglected.
>
> **Cop mechanism On Different Methods.** SubC: Subject Consistency; BkgC: Background Consistency; MoS: Motion Smoothness; AesQ: Aesthetic Quality; DyaD: Dynamic Degree.
>
> |                                 |       |       | CogVideoX |       |       |      |       |       | MotionClone |       |       |
> | :-----------------------------: | ----- | :---: | :-------: | ----- | ----- | ---- | ----- | ----- | :---------: | ----- | ----- |
> |                                 | SubC  | BkgC  |    Mos    | AesQ  | DyaD  |      | SubC  | BkgC  |     MoS     | AesQ  | DyaD  |
> |        All Coarse Prompt        | 0.853 | 0.892 |   0.765   | 0.512 | 0.734 |      | 0.712 | 0.823 |    0.657    | 0.425 | 0.689 |
> |       All detailed Prompt       | 0.911 | 0.912 |   0.662   | 0.511 | 0.501 |      | 0.789 | 0.871 |    0.881    | 0.662 | 0.842 |
> | with CoP Timestep Decomposition | **0.917** | **0.921** |   **0.893**   | **0.559** | **0.893** |      | **0.836** | **0.889** |    **0.912**    | **0.683** | **0.864** |
>
> To validate the design, we conducted two experiments:
>
> 1.  ablation study of using the same detailed prompt across all timesteps. The results demonstrate that our method yields superior consistency and fidelity, particularly in subject consistency and background consistency.
> 2. applying our chain-of-prompt strategy to recent VDM-based methods CogVideoX and MotionClone. We observed a notable performance improvement, indicating that CoP is also generalizable beyond our framework.
>    The quantitative results are demonstrated in the following tables. We will include these results and related discussions in the revised version of the paper.
>
> ​                                                                  **CoP mechanism with different level prompts.**
>
> |                     |       |       | Visual Quality |       |      |      | Video Smoothness |       |
> | :-----------------: | ----- | ----- | :------------: | ----- | ---- | ---- | :--------------: | ----- |
> |                     | LPIPS | PSNR  |      SSIM      | FID   | FVD  |      |       SubC       | BkgC  |
> |  All Coarse Prompt  | 0.174 | 23.15 |     0.701      | 72.46 | 433  |      |      0.714       | 0.865 |
> | All Detailed Prompt | 0.165 | 24.62 |     0.734      | 49.26 | 367  |      |      0.832       | 0.901 |
> |     Full Model      | **0.152** | **26.78** |     **0.803**      | **46.74** | **345**  |      |      **0.945**       | **0.931** |
>
> ---
>
> (3) Is it reasonable to understand 'Sec. 3.1.2 Self-supervised Pre-training via Random Masking' as finetuning the vdm to learn inpainting prior?
>
> **Answer**:  Yes, the random masking is utilized to finetune the vdm to learn inpainting prior to seamlessly fusing foreground and background without large-scale paired labels.
>
> ---
>
> (4) Is the model trained from scratch or finetuned from a pretrained video diffusion model? If the model is trained from scratch, why couldn't the authors finetune from a pretrained video model?
>
> **Answer:** The weights of the model are initialized from a pretrained CogVideoX-5B, which can leverage its generation ability. We will clarify this in the revision.
>
> ---
>
> (5) The paper omits discussing a highly relevant video concept personalization paper "Dynamic Concepts Personalization from Single Videos". What are the strength/limitation compared to this paper?
>
> **Answer:**
> 	*Dynamic Concepts Personalization* finetunes LoRA layers to learn an identity basis from videos that capture appearance while avoiding temporal interference, enabling realistic fusion effects through basis combination. Although it achieves realistic results, it requires separate training to fit for each scene and does not support target image input. In contrast, our feedforward approach allows for more efficient inference for all scenes. And we decouple the video generation into fore-ground image, background image and corresponding motions, making it more easier to transfer any of these parts. In addition, the two methods differ in focus: ours enables explicit subject transfer in video using a reference image, whereas theirs allows only limited object replacement via text prompt, making it limited application compared with us.
>
> ---
>
> (6) The paper omits discussing pioneering video motion transfer works, such as "MotionDirector: Motion Customization of Text-to-Video Diffusion Models", "Space-Time Diffusion Features for Zero-Shot Text-Driven Motion Transfer", and "VMC: Video Motion Customization using Temporal Attention Adaption for Text-to-Video Diffusion Models". Specifically, how does the method performs compared to MotionDirector?
>
> **Answer:**
> 	These three methods are all designed for text-guided video editing tasks, which lack the ability of pixel-level visual control. MotionDirector requires separate training for each motion type. When adapting to new scenarios, it only supports text-guided scene transitions, which significantly limits its flexibility. VMC also needs seperate training for different scenes.  Though 'Space-Time Diffusion Features for Zero-Shot Text-Driven Motion Transfer' is training free, when applied to motion-transfer, it only supports text guidance for new scene. In contrast, our method adopts both textural and visual guidance, enabling more precise control of the generated video. We will include the discussion in the related works of the revised version.

---

> ### Comment · Reviewer_Lpcd · 2025-08-04
> **Thank you for the rebuttal**
>
> The authors' rebuttal answers all my questions and I woud like to keep my rating.

---

> > ### Author Response · Authors · 2025-08-05
> >
> > Thanks for your feedback and for confirming that your questions have been addressed. We're glad to hear that. Please feel free to reach out if any further concerns arise.

---

### Official Review · Reviewer_FXM4 · 2025-07-03

**Clarity:** 2
**Significance:** 3
**Originality:** 2
**Rating:** 4
**Confidence:** 4

**Summary:**

This paper introduces UniTransfer, a novel architecture for precise and controllable video concept transfer. The core of UniTransfer lies in a progressive decomposition paradigm applied to both spatial and diffusion timestep dimensions. Spatially, it decouples a video into three key components: the foreground subject, the background, and the motion flow. This separation is managed by a dual-to-single-stream DiT-based architecture, enabling fine-grained control over each element.

**Questions:**

None

**Ethical Concerns:**

["NO or VERY MINOR ethics concerns only"]

**Final Justification:**

After reading other reviews, I keep the positive rating.

**Limitations:**

yes

**Quality:**

3

**Strengths And Weaknesses:**

Strength

Multi-stage Training Strategy: The model is first pre-trained on a large-scale dataset using a random masking strategy to acquire extensive prior knowledge. During this process, the model's learning is progressively guided by gradually increasing the size of the masked regions.

Granularity-aware Denoising: This work addresses the phenomenon in diffusion models where the denoising process first recovers structural information before progressively generating fine-grained textural details. To leverage this, a large language model is used to transform a single text prompt into three distinct, granularity-specific prompts. These are applied at different stages of the denoising process to focus the model's attention on the relevant aspects at each step.

Decomposition and Fusion for Versatility: The training strategy involves decomposing the video into foreground, background, and motion components, which are then fused. This framework allows the model to adapt to a wide variety of transfer and combination tasks, demonstrating strong versatility. In contrast, other methods are often limited to editing or transferring only single, more isolated regions or objects


Weakness

In the ablation study, was the comparison between the dual-stream and single-stream architectures conducted under the same computational cost? Similarly, was the comparison with early motion injection also performed under an equivalent computational budget?

Does the spatial branch use multi-frame noise? Why not use single-frame noise, given that this branch might only need to handle the spatial foreground-background relationship, while temporal dynamics are modeled by the motion branch? Modeling temporal features early on could potentially be misled by the text features.

Considering the results shown in Figure 4, Animate Anyone uses the pose sequence from the target video to drive a reference image. If we disregard the background and focus only on the foreground for comparison, the results in the second and third rows of Figure 4 suggest that the method proposed in this work is inferior to Animate Anyone in terms of preserving the person's appearance and identity.

This work uses optical flow as the motion condition. Does it account for the motion of the background? Generating consistent background motion would make the video more realistic. The framework could be further improved by considering the motion of multiple objects.

---

> ### Author Rebuttal · Authors · 2025-07-31
>
> (1) In the ablation study, was the comparison between the dual-stream and single-stream architectures conducted under the same computational cost? Similarly, was the comparison with early motion injection also performed under an equivalent computational budget?
>
> **Answer：**
> Thank you for your thoughtful questions. Here is our response regarding the two points:
>
> - **Comparison between dual-stream and single-stream architectures.**
>   Our current experiments are built upon the pretrained CogVideoX-5B backbone. The single-stream model also maintains a 5B-scale parameter count, taking about $180$ seconds to generate a $720\times480$ video with $49$ frames with an Nvidia-A100 GPU. However, the dual-stream architecture includes an additional foreground branch, which increases the total parameter count by approximately 30% and the inference time by about 19\%. To ensure a fair comparison under an equivalent computational budget, we also reconstructed the dual-stream model using a smaller CogVideoX-2B backbone, resulting in a total model size of around 3B parameters. Despite the smaller size, this 3B dual-stream model outperforms the original 5B single-stream baseline in our evaluations, as shown in the following table. This result provides further evidence of the architectural effectiveness of our dual-stream design. We will include this additional comparison in the revised version and add more qualitative results to guarantee the fairness of the ablation setup.
>
> - **Comparison with early motion injection.**
>   The comparison with early motion injection was conducted under an equivalent computational budget, using the same base model and training settings. The only difference is the injection position of motion features, ensuring that the performance gains are due to the proposed motion-injection strategy, not model capacity or training cost differences.
>
> **Comparison of different base architectures.** SubC: Subject Consistency; BkgC: Background Consistency; MoS: Motion Smoothness; AesQ: Aesthetic Quality; DyaD: Dynamic Degree.
> |                       | SubC  | BkgC  |  MoS  | AesQ  | DyaD  |
> | :-------------------: | :---: | :---: | :---: | :---: | :---: |
> |   Single-stream-5B    | 0.927 | 0.904 | 0.926 | 0.557 | 0.842 |
> |  Ours-Dual-stream-3B  | 0.933 | 0.913 | 0.933 | 0.591 | 0.869 |
> | Ours-Dual-stream-6.5B | **0.945** | **0.931** | **0.962** | **0.651** | **0.903** |
>
> ---
>
> (2) Does the spatial branch use multi-frame noise? Why not use single-frame noise, given that this branch might only need to handle the spatial foreground-background relationship, while temporal dynamics are modeled by the motion branch? Modeling temporal features early on could potentially be misled by the text features.
>
> **Answer:**
> 	Our model is built upon the CogVideoX architecture, and we follow its design paradigm by adopting multi-frame noise as input across all branches, including the spatial branch. While the motion branch primarily models temporal dynamics, the spatial branch in our architecture is mainly responsible for integrating foreground-background structures. However, the input motions may not always align perfectly with the foreground or background. In these cases, dynamics of the unaligned areas can be generated with the guidance of text prompts by introducing multi-frame noise in the spatial branch and enabling coarse temporal modeling during the early stage. This helps to suppress artifacts that may be caused by misaligned motion signals. We recognize the reviewer's concern that early modeling of temporal features could be misled by text prompts. In our work, during the initial denoising steps, only coarse textual guidance is provided. Finer-grained prompts are introduced progressively as the denoising proceeds. This staged guidance reduces the risk of misleading caused by inappropriate textual descriptions.
>
> ---
>
> (3) Considering the results shown in Figure 4, Animate Anyone uses the pose sequence from the target video to drive a reference image. If we disregard the background and focus only on the foreground for comparison, the results in the second and third rows of Figure 4 suggest that the method proposed in this work is inferior to Animate Anyone in terms of preserving the person's appearance and identity.
>
> **Answer：**
> 	We would like to clarify that our work focuses on video concept transfer, where the goal is to substitute the target concept in the video with a user-specified new concept, while preserving the remaining video content. AnimateAnyone is designed specifically for pose-driven human video generation from a single reference image. It assumes that the appearance and identity of the subject should remain fixed. While AnimateAnyone may appear to preserve the details better in certain examples, our solution significantly outperforms it in terms of overall object motion coherence and background consistency. Please refer to the videos on our anonymous webpage given in the paper for a holistic and objective comparison.
>
>
> ---
>
> (4) This work uses optical flow as the motion condition. Does it account for the motion of the background? Generating consistent background motion would make the video more realistic. The framework could be further improved by considering the motion of multiple objects.
>
> **Answer：**
> Thank you for your constructive suggestions regarding motion modeling for background and multiple objects.
>
> - **Background Motion Handling.**
>
>   We agree that generating consistent background motion can enhance video realism. Our current framework is capable of handling small-scale background motion—see the middle video in the second row of "Human foreground transfer" on our anonymous site (given in the paper), where global camera movements are reflected in generated videos. However, large-scale or complex background motion is challenging for our method. As the reviewer suggested, incorporating explicit background motion generation could further improve realism, and we consider this a promising direction for future work. We plan to explore learning or estimating background motion patterns jointly with foreground objects to achieve a more holistic motion representation.
>
> - **Multi-Object Motion Modeling.**
>   For scenarios involving multiple independently moving objects, we plan to fuse their individual optical flows into a composite representation and sequentially condition the network on each object through multiple inference passes. We will include corresponding results in the revised version of the paper.

---

### Decision · Program_Chairs · 2025-09-17

**Decision:**

Accept (poster)

**Comment:**

This paper was reviewed by four experts in the field. Most concerns are solved in the rebuttal and discussions. The paper finally received positive reviews with 2 Borderline Accepts and 2 Accepts.

This paper proposes UniTransfer, a framework controllable video concept transfer. It spatially decomposes videos into foreground, background, and motion flow, using a dual-to-single-stream DiT architecture for fine-grained control. It employs a chain-of-prompt strategy for timestep decomposition, leveraging LLMs to guide the denoising process progressively.

**strengths**
- the dual-decomposition strategy is technically sound and motivated, e.g., spatial disentanglement, motion conditioning, progressive textual control, etc.
- a unified framework that handles character, garment, background and motion transfer in a single model.
- comprehensive experimental results across several tasks

**weaknesses**
- cannot handle large-scale or complex background motion, complex interactions
- complicated framework (three-branch DiT, a LLM, and optical-flow estimator), which might introduce non-determinism and complexity

According to the reviews, the reviewers acknowledge the paper's novel design of the decomposition strategy as well as the versatility of the unified framework.
Following the rebuttal, all reviewers have provided positive ratings for the paper.
In the revision, the authors are suggested to incorporate the reviewers' feedback, such as the implementation details, the running time and memory usage, and the required experimental results.